# Genome-Wide Identification and Expression Analysis of Beta-Galactosidase Family Members in Chinese Bayberry (*Myrica rubra*)

Li Sun [1,2,*,†], Qinpei Yu [1,3,†], Shuwen Zhang [1,2], Zheping Yu [1,2], Senmiao Liang [1,2], Xiliang Zheng [1,2], Haiying Ren [1,2] and Xingjiang Qi [1,2,4,*]

1 Institute of Horticulture, Zhejiang Academy of Agricultural Sciences, Hangzhou 310021, China; yuqinpei0614@163.com (Q.Y.); zhangsw@zaas.ac.cn (S.Z.); yuzp@zaas.ac.cn (Z.Y.); liangsm@zaas.ac.cn (S.L.); zhengxl@zaas.ac.cn (X.Z.); renhy@zaas.ac.cn (H.R.)
2 State Key Laboratory for Managing Biotic and Chemical Threats to Quality and Safety of Agro-Products, Hangzhou 310021, China
3 College of Life Sciences, Zhejiang Normal University, Jinhua 321004, China
4 Xianghu Laboratory, Hangzhou 311231, China
* Correspondence: sunli@zaas.ac.cn (L.S.); qixj@zaas.ac.cn (X.Q.)
† These authors contributed equally to this work.

**Abstract:** Fruit development and softening play pivotal roles in determining fruit quality and post-harvest shelf life in Chinese bayberry (*Myrica rubra*). However, the specific role of beta (β)-galactosidase, particularly β-galactosidase of *M. rubra* (*MrBGAL*), in facilitating fruit softening remains unclear. In this study, we aimed to address this gap by investigating the involvement of *MrBGALs* genes in fruit softening. We identified all 15 *MrBGALs* and conducted a comprehensive analysis, including phylogenetic relationships, gene structure, protein motifs, co-linearity, and expression patterns. Using phylogenetic analysis, we classified all *MrBGALs* into five distinct groups. Additionally, cis-element prediction and comparative genome analysis provided insightful clues about the functionality of *MrBGALs*. Transcriptome data revealed unique expression patterns of *MrBGALs* throughout various fruit development stages. These findings introduce valuable candidate genes that can contribute to unraveling the functions and molecular mechanisms governing fruit development and softening in Chinese bayberry.

**Keywords:** *Myrica rubra*; β-galactosidase family; transcriptome; fruit development

## 1. Introduction

Chinese bayberry (*Myrica rubra* Sieb. et Zucc.) belongs to the *Myricaceae* family, with *Myrica* as its predominant genus, comprising about 50 species [1]. It is a subtropical plant native to China that has been cultured for thousands of years in China and other Asian countries [2]. Recognized for its delicious taste and versatility, Chinese bayberry's fruits are consumed fresh or undergo processing into diverse products such as juice, jam, sweets, and wine. Beyond their culinary appeal, the fruits are acclaimed for their health benefits attributed to antioxidants, showcasing anti-inflammatory [3,4], anti-diabetic [5,6], and anti-bacterial [4,7] properties. The ripening period of Chinese bayberry is brief, aligning with the plum rain season and post-ripening, the fruit undergoes rapid softening, leading to significant damage and substantial economic losses. Consequently, the promotion of slow-softening traits and the cultivation of new varieties with early maturation characteristics have emerged as crucial factors for the sustainable advancement of this industry.

Fruit softening is a critical determinant of fruit quality and post-harvest shelf life, representing a pivotal aspect of the ripening syndrome. It involves biochemical changes in cell wall fractions that are associated with the breakdown of cell wall polymers [8,9]. This process is catalyzed by cell wall hydrolytic enzymes, including polygalacturonase,

pectin methylesterase, pectate lyase, and β-galactosidase [10–12]. Among these enzymes, polygalacturonase [13] and β-galactosidase [14,15] have been extensively studied and are considered the most potent enzymes regulating fruit softening. Plant β-galactosidase, a constituent of the glycoside hydrolase 35 (GH35) families [16], plays a crucial role in catalyzing the removal of terminal galactosyl residues from carbohydrates, glycoproteins, and galactolipids [17]. The β-galactosidase family (BGAL) has been identified in various plant species, such as Arabidopsis [18], tomato [19] and sweet potato [20]. The GH35 conserved domain of the BGAL in the N-terminal region [21] is typically responsible for the hydrolytic activity of β-galactosidase. The protein subcellular location of the BGAL in melon and Arabidopsis showed that they were mainly located in extracellular space (cell wall) [11]. The BGAL play significant roles throughout a plant's lifecycle, influencing plant growth and development, including fruit softening [22–26], fruit development and ripening [19,27,28], seed germination [29], stress responses [30,31], and root development [32]. Furthermore, β-galactosidase is believed to expedite fruit softening [33].

β-galactosidase is believed to expedite fruit softening by increasing the porosity of the cell wall, thereby enhancing the access of other cell wall-degrading enzymes [34]. The activity of β-galactosidase remained stable within the pH range of 3.5–8.5, with the highest activity observed at pH 4.5 in Arabidopsis [18,35]. Furthermore, Arabidopsis *BGAL4* was stable up to 45 °C but lost activity precipitously at higher temperatures, suggesting that β-galactosidase maintains relatively high activity across various fruit-ripening stages. Seven BGALs in tomato are reportedly expressed throughout fruit development, with six identified as expressed during ripening. The downregulation of tomato β-galactosidase 4 (TBG4) has been linked to reduced fruit softening [36]. Moreover, suppressing β-galactosidase activity in the early stages of ripening markedly diminishes fruit softening [9], indicating the important functions of the BGAL as crucial contributors to cell wall changes associated with ripening-related firmness loss. *Mdβ-GAL18* a pectin-degrading enzyme associated with cell wall metabolism was upregulated via exogenous ethylene treatment. In apples, ethylene-induced MdZF-HD11 interacts with *Mdβ-GAL18* to promote the post-harvest softening of apples [22]. The use of a specialized universal primer facilitates a comprehensive understanding of the expression patterns of the BGAL family genes [37]. Understanding the global landscape of BGAL genes in Chinese bayberry (*MrBGALs*) is crucial for the cultivation of slow-softening Chinese bayberry.

While the BGAL have been extensively identified in various plant species [38], their function in Chinese bayberry remains elusive. In this study, we have, for the first time, identified 15 *MrBGALs*. Subsequently, we explored their phylogeny, motif compositions, and predicted cis-elements using diverse bioinformatics tools. Our investigation aims to establish the groundwork for comprehending the role of *MrBGALs* through a comprehensive analysis, providing innovative insights into the regulatory mechanisms of *MrBGALs* influencing plant growth.

## 2. Methods

### 2.1. Identification of MrBGALs

The PF01301 domain, cataloged in the Pfam database, has been recurrently linked with β-galactosidase enzymes, indicating a probable involvement in shaping their structure or influencing their function. In our quest to pinpoint *MrBGALs* (putative β-galactosidases), we employed the hmmscan tool to meticulously filter through candidate sequences harboring the PF01301 domain. Our set inclusion criteria were meticulously defined: a full sequence E-value less than $1 \times 10^{-5}$, a domain i-Evalue less than $1 \times 10^{-5}$, and a minimum coverage of the PF01301 domain in the target sequence equal to or exceeding 80%.

### 2.2. Analysis of Co-Linearity

Co-linearity analysis was conducted through a systematic series of steps. Initially, the protein sequences of Chinese bayberry were aligned using a self-to-self comparison facilitated using BLAST software. This alignment process was executed with specific

parameters, including an E-value threshold of $1 \times 10^{-5}$, a tabular output format (-m8), settings for verbosity (-v) set at 5, and number of database sequences to show (-b) set at 5. Following sequence alignment, the identification of co-linearity relationships was accomplished using MCScanX software. Subsequently, the classification of *MrBGALs* (putative β-galactosidases) was carried out using the duplicate_gene_classifier module of MCScanX to categorize the identified genes based on their duplications and potential functional significance [39].

### 2.3. The Phylogenetic Tree, Gene Structures, and Motif Analysis of MrBGALs

To delve into the phylogenetic relationships among MrBGALs, we initiated the process by extracting amino acid sequences featuring the conserved Glyco_hydro_35 domain (PF01301). These sequences were then subjected to alignment using MUSCLE software v3.8.1551 [40] and subsequently constructed an evolutionary tree through FastTree 2.1.11 [41]. The resulting evolutionary tree, along with the gene structure diagram, was visualized using TBtools v1 [42]. Additionally, we conducted a search for conserved motifs using MEME 5.0.5 [43].

### 2.4. Cis-Element Prediction

To anticipate the putative cis-elements governing the expression of *MrBGALs*, we began by extracting upstream promoter sequences, spanning a region of 1000 base pairs (bp) upstream of the transcription start site. Subsequently, these extracted promoter sequences were subjected to a meticulous cis-element prediction analysis employing the widely recognized bioinformatics tool, PlantCARE (https://bioinformatics.psb.ugent.be/webtools/plantcare/html/, accessed on 24 January 2024). The PlantCARE database specializes in the identification of cis-regulatory elements in plant promoters.

### 2.5. Genomes for Evolutionary and Comparative Genome Analysis

The genetic information was acquired from the Ensembl database (https://plants.ensembl.org, accessed on 24 January 2024). The five genomes retrieved comprised *Solanum lycopersicum* (SL3.0), *Cucumis melo* L. (Melonv4), *Arabidopsis thaliana* (L.) (TAIR10), *Prunus persica* (PAV_r1.0), and *Malus domestica* (ASM211411v1). Additionally, the genome of *Fragaria vesca* L. was obtained from http://eplant.njau.edu.cn/strawberry/, accessed on 24 January 2024, and the genome of *M. rubra* was downloaded from http://cotton.zju.edu.cn/source/Myrica_rubra.zip, accessed on 24 January 2024 [44].

### 2.6. RNA Sequencing

Total RNA was extracted from three stages—unripe (UR), middle ripe (MR), and full ripe (FR)—of *M. rubra* Zaojia fruit using TRIzol reagent (Invitrogen, Waltham, MA, USA), and its integrity was evaluated through RNA integrity numbers using the Agilent 2100 (Santa Clara, CA, USA) system, following the manufacturer's guidelines. Library construction was executed using an NEB Next Ultra II RNA Library Prep Kit. Subsequently, sequencing was conducted on an Illumina HiSeq X Ten machine in pair-end mode, generating 150 base pairs per read (Genewiz, South Plainfield, NJ, USA).

### 2.7. Analysis of RNA Sequencing Data

RNA sequencing data were analyzed as previously described [45]. Briefly, the sequencing reads underwent trimming and alignment to the reference genome of Zaojia using Fastp v0.23.2 [46] and HISAT2 v2.2.1 [47], respectively. FPKM (fragments per kilobase of transcript per million mapped reads) values were computed utilizing StringTie v1.3.3b. Differentially expressed genes were identified employing the R package Ballgown, applying cutoff criteria of FPKM fold change >1.5 and a *p*-value of <0.05 [48].

## 3. Results

### 3.1. Identification and Characterization of MrBGALs

To systematically identify and characterize *MrBGALs*, we extracted the longest protein coding sequences, subjected them to annotation using hmmscan, and applied stringent criteria to filter candidate sequences containing the PF01301 domain. A total of 15 candidate *MrBGALs* met these criteria (Supplementary Table S1). The distribution of *MrBGALs* across chromosomes revealed an uneven pattern, with representations on Chr1, Chr3, Chr5, Chr6, Chr7, and Chr8. Notably, certain genes, exemplified by *MrChr3G28670* and *MrChr3G28800*, were found in close proximity to each other on a single chromosome. This spatial arrangement suggests a potential functional collaboration between these genes or a possibility of duplicated functions. To further elucidate the genomic organization and evolutionary patterns, we conducted synteny analysis for all these *MrBGALs*. The results revealed that 5 gene pairs from 15 *MrBGALs* appeared to have stemmed from whole-genome duplication (WGD) or segmental duplications. These gene pairs were distributed on four Chinese bayberry chromosomes and most frequently on Chr7 and Chr 8. Meanwhile, three GT1 genes were likely to be tandem duplicates, suggesting that WGD or segmental duplication and tandem duplication played comparably important roles in the evolution of these *MrBGALs* (Figure 1A, Supplementary Table S2).

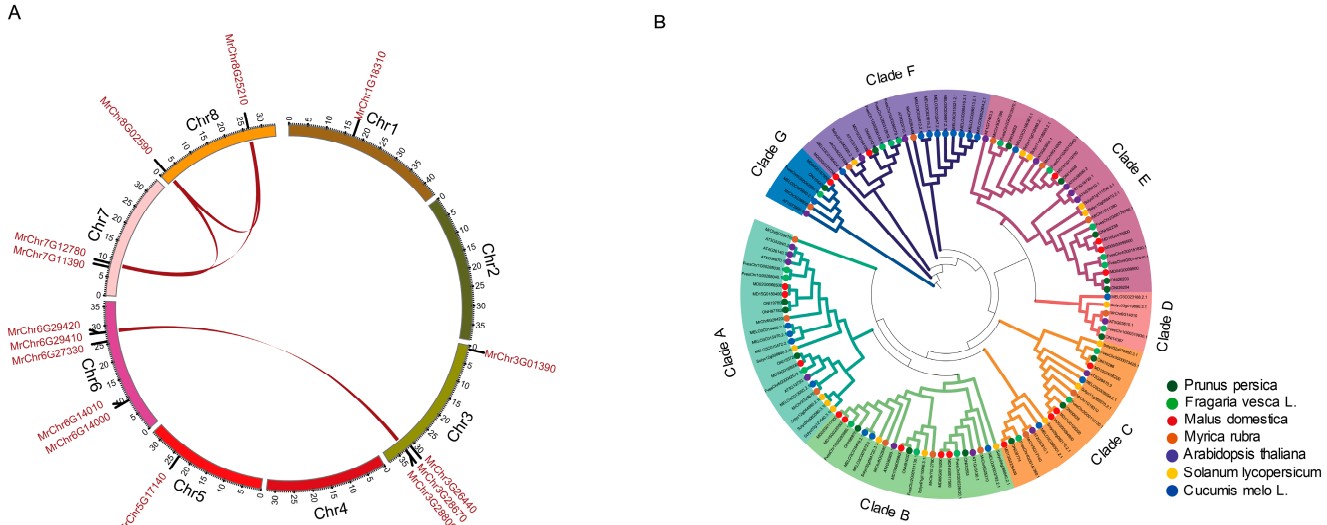

**Figure 1.** Identification and evolutionary analysis of *MrBGALs*. (**A**) Circos plot of *M. rubra* genome annotated with *BGAL* genes. The eight chromosomes are arranged circularly and represented by labeled ideograms. On the outside of the ideograms, the *BGAL* genes' accession numbers were labeled based on their chromosomal position. Links with dark red color show the segmental duplication or WGD relationship between *BGAL* genes. (**B**) Phylogenetic tree of *BGAL* genes among *Myrica rubra*, *Cucumis melo*, *Arabidopsis thaliana*, *Solanum lycopersicum*, *Prunus persica*, *Malus domestica*, and *Fragaria vesca*. The gene accession numbers of all the *BGAL* genes are shown in Table S1.

### 3.2. Evolutionary Analysis of MrBGALs and Other Plants

To enhance the contextualization of the evolutionary classification, we applied the analytical approach previously used for the identification of *MrBGALs* to extract the longest transcripts from the selected six species for evolutionary analysis. Individually scrutinizing the BGAL family members of *Arabidopsis thaliana* (L.) *Heynh*, *Solanum Lycopersicum*, *Cucumis melo*, *Prunus persica*, *Fragaria vesca*, and *Malus domestica*, we aligned the protein sequences of family members and constructed an evolutionary analysis, followed by clade classification. Our comprehensive analysis revealed that the 122 *BGALs* from 7 species could be systematically categorized into 7 groups: Clade A (24), Clade B (22), Clade C (19), Clade D (6), Clade E (27), Clade F (20), and Clade G (6) (Figure 1B). Genes within the same clade exhibit a notable potential for shared functionality, thereby allowing us to infer and predict the func-

tional attributes of *MrBGALs* based on their placement within corresponding branches. The expression levels of *Mdβ-Gal1* (*MD09G0168900*, Clade A), *Mdβ-Gal3* (*MD02G0117600*, Clade B), *Mdβ-Gal5* (*MD10G0056800*, Clade B), and *Mdβ-Gal11* (*MD10G0078800*, Clade B) gradually increased during apple fruit development [28]. This observation suggests a potential association of these clades (Clade A: *MrChr6G29410*, *MrChr6G29420*, and *MrChr3G28670*; Clade B: *MrChr8G02590*, *MrChr7G12780*, and *MrChr8G25210*) with fruit development.

### 3.3. Analysis of Gene Structures and the Motif Composition Conserved Domain of MrBGALs

In our investigation of *MrBGAL* structures, we aligned the sequences of 15 *MrBGALs* and constructed an evolutionary tree, providing a coherent illustration of their evolutionary relationships. The categorization of these 15 *MrBGALs* unveiled seven distinct groups (Figure 2A). Notably, Group A, Group B, and Group E emerged as the predominant clusters, each accommodating three genes. In contrast, Group C and Group F encompassed two genes each, while Group D and Group G comprised a singular gene. Group G stood out notably from the others in the evolutionary tree, exhibiting marked differences, indicating that its functionalities may differ from those of the other groups. An analysis of conserved motifs further refined our classification. Ten motifs were identified across these groups (Supplementary Table S3). Groups A, B, C, D, and F encompassed all ten motifs, whereas Group G contained only Motif1 and Motif2. Group E stood out by lacking Motif8 (Figure 2B). Upon scrutinizing the protein domain composition, it was evident that all *MrBGALs* harbor the functional Glyco_hydro_35 domain, the primary β-galactosidase active domain within the plant glycoside hydrolase family 35. However, the presence of the GHD and galactose-adhering lectin (Gal_lectin) domains is gene-specific. All *MrBGALs*, except *MrChr3G28800*, feature the GHD domain. In contrast, *MrChr3G01390*, *MrChr6G14000*, *MrChr7G12780*, *MrChr6G29410*, and *MrChr3G28800* lack the galactose-adhering lectin domain, and the remaining genes possess Gal_lectin domains at the C terminus of the protein sequence (Figure 2C). These results suggest that all *MrBGALs* demonstrated β-galactosidase activity, indicating their potential role in augmenting the porosity of the cell wall. This, in turn, could enhance the accessibility of other cell wall-degrading enzymes, consequently influencing fruit development and softening processes.

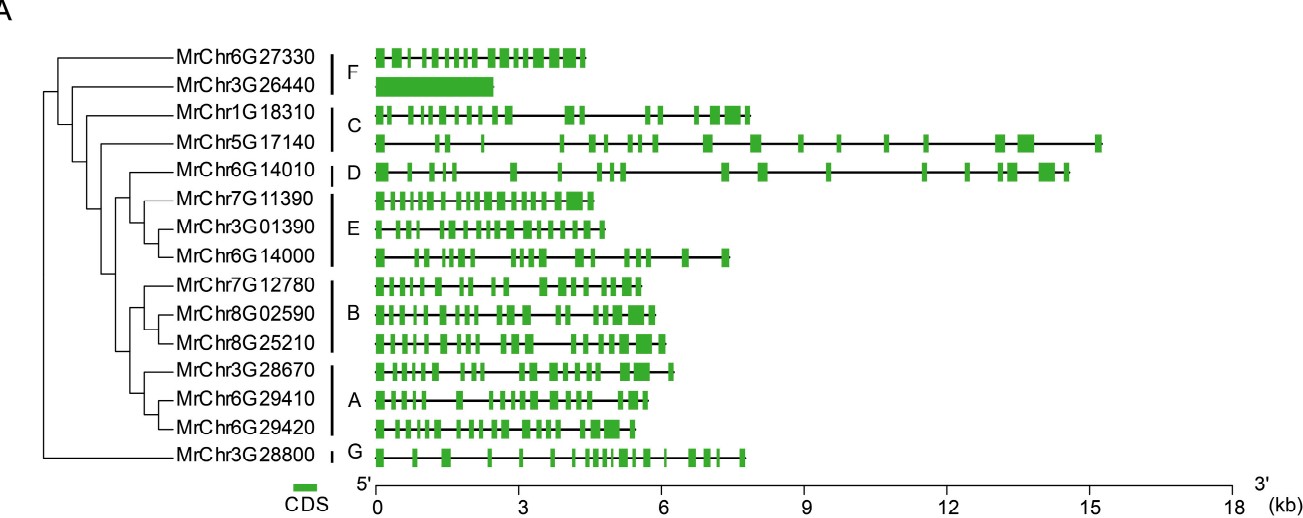

**Figure 2.** *Cont.*

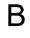

B

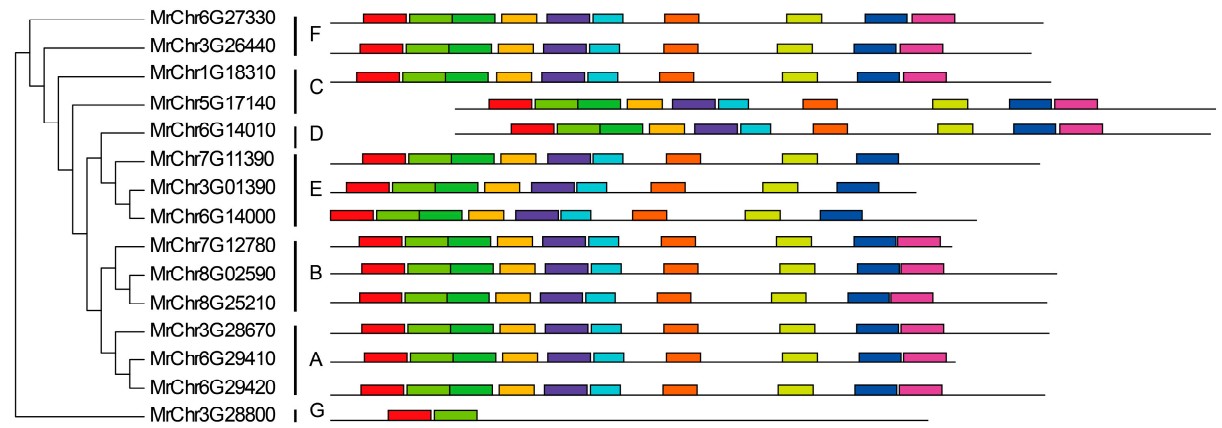

| | Consensus Sequence | E-value | Width (aa) |
|---|---|---|---|
| Motif 1 | IIBGQRRILISGSIHYPRSTPEMWPDLIQKAKEGGLDVIZTYVFWNLHEP | $1.5 \times 10^{-505}$ | 50 |
| Motif 2 | GGPFITTSYDYDAPJDEYGLJRQPKWGHLKELHAA | $3.3 \times 10^{-307}$ | 35 |
| Motif 3 | YNFEGRYDLVKFIKLVQKAGLYVHLRIGPYVCAEWNFGGFPVWLKYVPGI | $5.2 \times 10^{-438}$ | 50 |
| Motif 4 | WTENWTGWFTEFGGPPPHRPAEDJAFAVARFIQKGGSFVNYYMYHGGTNF | $1.4 \times 10^{-43}$ | 50 |
| Motif 5 | AYVRWAAKMAVGLGTGVPWVMCKQKDAPDPVINTCNGFYCD | $7.7 \times 10^{-390}$ | 41 |
| Motif 6 | VFRTDNEPFKYEMQKFTAKIVDMMKEERLFASQGGPIILSQIENEYGPVE | $6.0 \times 10^{-406}$ | 50 |
| Motif 7 | KQQPLTWYKTYFDAPAGNDPVALDMGGMGKGQAWVNGQSIGRYWPAYAA | $5.3 \times 10^{-349}$ | 40 |
| Motif 8 | CNECSYRGTYNPKKCLTNCGEPSQRWYHVPRSWLKPSGNLLVVFEELGG | $1.7 \times 10^{-274}$ | 50 |
| Motif 9 | FNNGSYHLPPWSISILPDCKNVVFNTAKVGTQTSQMKMKP | $4.0 \times 10^{-238}$ | 40 |
| Motif 10 | TFEKPVSLKAGVNKISLLSVTVGLPNYGAFFETWNAGILGP | $5.0 \times 10^{-167}$ | 41 |

C

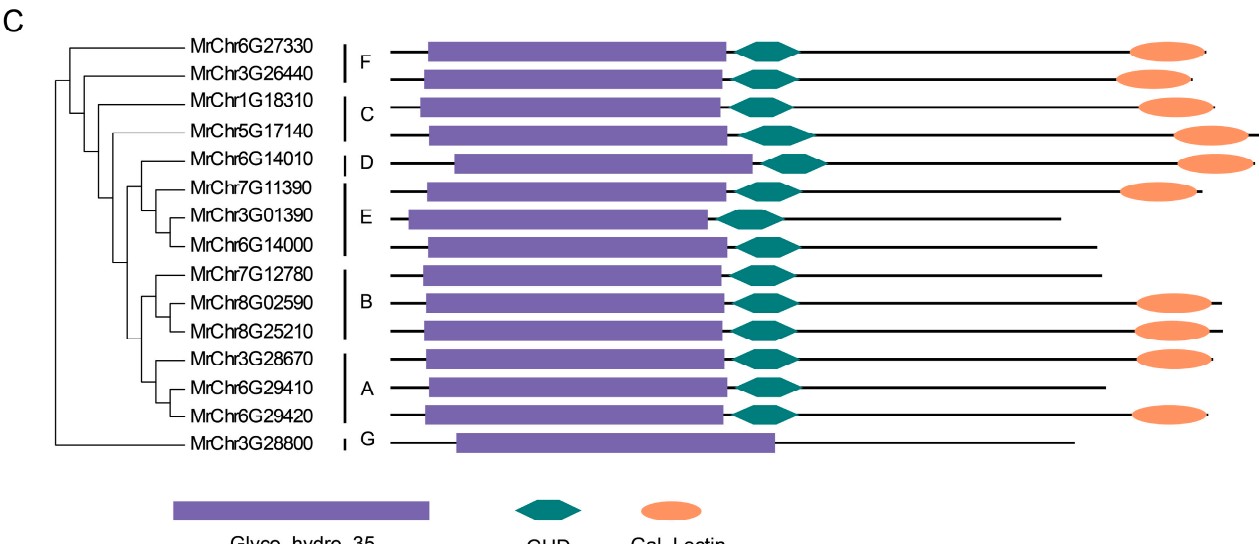

**Figure 2.** Phylogenetic tree, gene structure, and motif analysis of *MrBGALs*. (**A**) Phylogenetic tree was clustered into seven clades: A, B, C, D, E, F, G (left). Gene structure of *BGAL* genes and CDS regions are displayed with green filled rectangles, and the black line represent the intron regions (right). (**B**) Phylogenetic tree of *BGAL* genes (left). Distribution of conserved motifs of top 10 along the gene regions of BGAL genes (right). The sequences and the lengths of top 10 conserved motifs filtered with the e-value $1 \times 10^{-5}$ (bottom). (**C**) Phylogenetic tree with conserved domains. Phylogenetic tree of BGAL genes (left). Conserved domains' distribution (right).

### 3.4. Cis-Element Prediction of MrBGALs

Promoters serve as pivotal regulators of gene expression, exerting influence over both temporal and spatial dimensions through interactions with specific cis-regulatory elements. These cis-elements, short DNA sequences, assume a critical role in orchestrating gene expression by engaging with transcription factors and other regulatory proteins. To elucidate the potential upstream transcriptional regulatory mechanisms of *MrBGALs*, we

predicted and analyzed the cis-elements within the promoter sequences of each *MrBGAL* using PlantCARE. A total of 13 cis-elements were identified in the promoters of *MrB-GALs*. Most *MrBGALs* promoters had the GARE (gibberellin-responsive element), AuxRE (Auxin-responsive element) and ABRE (abscisic acid-responsive element) motifs which were involved in plant hormone response (Figure 3). Gibberellins, auxins, and abscisic acid play a crucial role in regulating various aspects of plant growth and development, including fruit development and softening. The association between cis-regulatory elements and pathways, as well as the quantity of cis-regulatory elements associated with each pathway for *MrBGALs*, suggests that diverse plant hormone-responsive elements, including those for auxin, abscisic acid, salicylic acid, gibberellin, and methyl jasmonate (MeJA), may mediate the expression of *MrBGALs* (Supplementary Tables S4 and S5), which suggests that *MrB-GALs* may constitute a significant component within the regulatory system governing fruit development and softening. Furthermore, the presence of light-responsive elements, stress-related elements (such as those associated with low temperature and drought), and other regulatory motifs indicates the potential responsiveness of *MrBGALs* to these inductions.

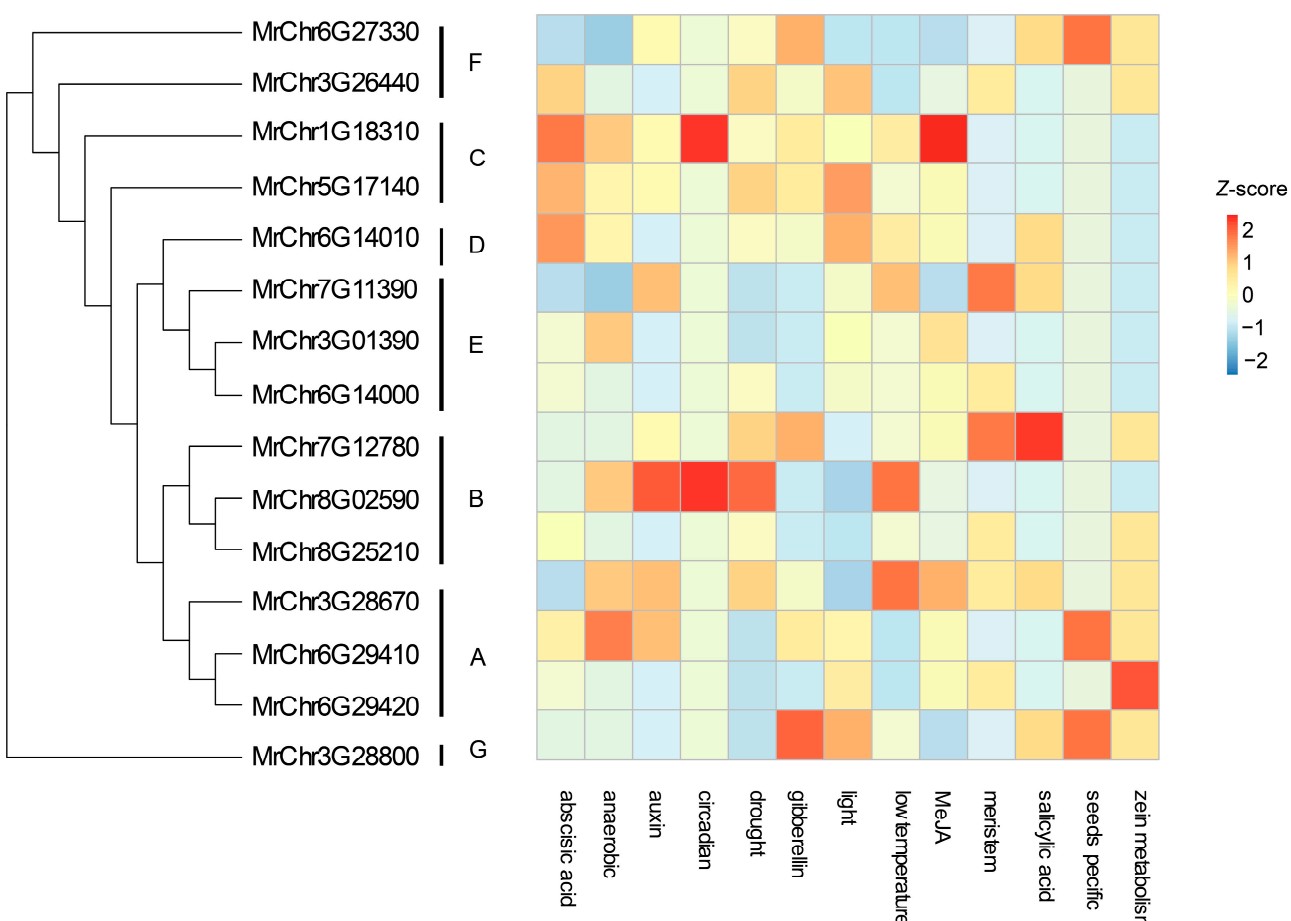

**Figure 3.** Phylogenetic tree and cis-regulatory element analysis. Phylogenetic tree of *BGAL* genes (**left**). Heatmap of the number of cis-regulatory elements, which were classified by pathways (**right**); scale unit is Z-score.

### 3.5. Comparative Genome Analysis between Arabidopsis thaliana, M. rubra, and Solanum lycopersicum

To further explore the phylogenetic mechanism of *MrBGALs*, comparative syntenic maps between *M. rubra* and the two representative species, *Arabidopsis thaliana* and *Solanum lycopersicum*, were constructed. By integrating the multi-species evolutionary tree of the BGAL family, we identified 15 orthologous gene pairs exhibiting co-linearity between

*Arabidopsis thaliana* and *M. rubra*, while 10 orthologous gene pairs exhibed co-linearity between *M. rubra* and *Solanum lycopersicum* (Figure 4, Supplementary Table S6), demonstrating a noteworthy degree of conservation. While *Gal4* in tomato and strawberry has been correlated with reduced fruit softening [24,36], our investigation revealed that *Atβ-Gal4* (*AT5G56870.1*) and *MrChr6G29420* formed a gene pair, implying that they have a potential role in the fruit softening process of *MrChr6G29420*.

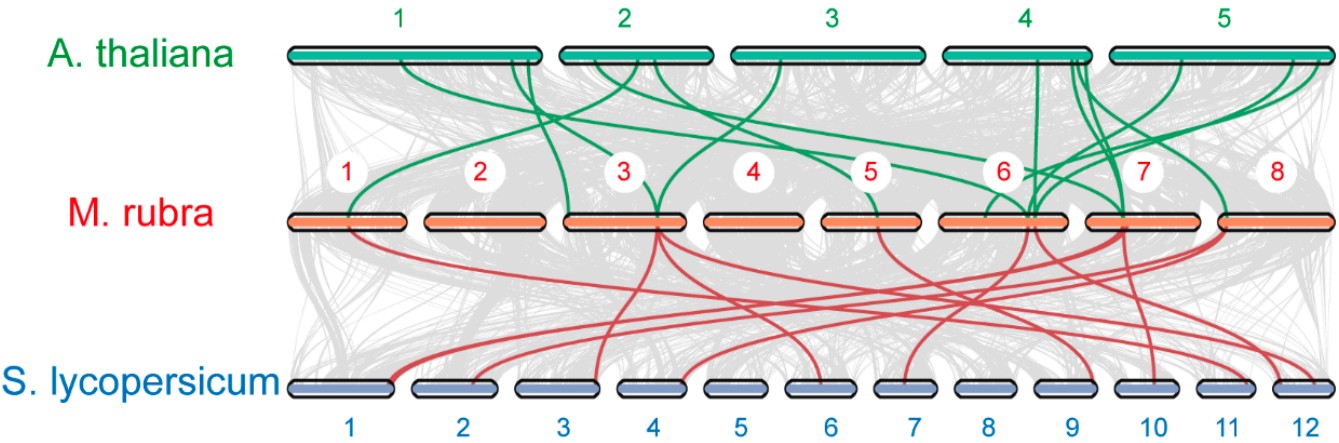

**Figure 4.** Co-linearity between 5 chromosomes of *Arabidopsis thaliana*, 8 chromosomes of *Myrica rubra*, and 12 chromosomes of *Solanum lycopersicum*. The orthologous genes between *Arabidopsis thaliana* and *Myrica rubra* are highlighted with a green color, and the orthologous genes between *Myrica rubra* and *Solanum lycopersicum* are presented with a red color.

*3.6. Expression Profiles in Different Fruit Development Stages*

To elucidate the potential functions of *MrBGALs* genes, we analyzed transcriptome data from three distinct developmental stages—namely, unripe (UR), middle ripe (MR), and full ripe (FR)—of ZaoJia, a subspecies of Chinese bayberry. In our findings, 10 out of 15 *MrBGALs* genes were identified in this subspecies. Compared to the UR stage, 2183 genes were upregulated, and 2294 genes were downregulated in the MR stage, with 8 out of 10 *MrBGALs* displaying upregulation (Figure 5A, Supplementary Table S7). During the UR to MR period, the initiation of fruit development occurs, necessitating substantial nutritional resources for growth, with the upregulation of a majority of *MrBGALs* facilitating the developmental processes leading to the formation of a firm fruit. Compared to the UR stage, 3526 genes were upregulated, and 3621 genes were downregulated in the FR stage, with 6 out of 10 *MrBGALs* exhibiting upregulation (Figure 5B, Supplementary Table S8). Throughout the entire ripening process, the prevalent upregulation of the majority of *MrBGALs* implies their active involvement in the processes of fruit development and softening. Interestingly, compared to the MR stage, the FR stage revealed 1819 upregulated genes and 2136 downregulated genes, with 2 out of 10 *MrBGALs* being significantly downregulated (Figure 5C, Supplementary Table S9). In this post-fruit-ripening stage, *MrBGALs* experience downregulation, signifying a distinct regulatory pattern compared to the initiation of fruit development, which may be associated with the post-development and softening of the fruit. These results underscore the crucial role of *MrBGALs* across all fruit development stages. Notably, the majority of *MrBGALs* exhibited upregulation during the early stage of fruit ripening, transitioning towards either no change or downregulation in the post-fruit-ripening stage. This pattern indicates a distinctive up-to-down modulation in the fate of *MrBGALs* during fruit ripening. *MrChr6G29420* displayed notable upregulation in both the unripe/middle and middle/full ripening stages, suggesting an active function throughout the entire stage. This observation implies a potential association with fruit development and softening processes. To further understand the expression pattern of *MrBGALs*, we compared all annotated *MrBGALs* across the three stages, revealing a consistent pattern of first rise then descend (Figure 5D). Subsequently, we performed gene ontology (GO)

enrichment analysis utilizing differentially expressed genes. Notably, during the UR-MR stage, two of the differentially expressed *MrBGALs*, *MrChr6G29420* and *MrChr3G28670*, were identified as intracellular components, which indicates that these two genes play pivotal roles in the initiation stage of fruit development and that their upregulation may be induced by the onset of fruit ripening. In the subsequent UR-FR stage, *MrChr3G28670* was associated with cellular anatomical entities, demonstrating the significance of this gene throughout the entire stage of fruit development. Moreover, during the MR-FR stage, *MrChr5G17140* was identified in relation to cellular anatomical entities as well, suggesting that it could serve as a supplementary support for *MrChr3G28670* during the post-fruit-ripening stage (Figure 5E). These findings indicate diverse roles of *MrBGALs* in various stages and aspects of fruit development, ranging from intracellular processes to extracellular functions. Additionally, during the UR-MR stage, three *MrBGALs* were enriched in the molecular function of binding, indicating that their potential role may be activated by cis-regulation and that these genes could functionally act as regulators in the downstream regulation system (Figure 5F). Furthermore, KEGG enrichment analysis for differentially expressed genes in the three stages revealed significant enrichment only in the UR-MR stage, particularly in glycosphingolipid biosynthesis and glycosaminoglycan degradation, which are both associated with fruit softening (Figure 5G). In this pathway, the following seven out of ten *MrBGALs* participated in these pathways: *MrChr3G28670*, *MrChr3G28800*, *MrChr5G17140*, *MrChr6G29410*, *MrChr6G29420*, *MrChr8G02590*, and *MrChr8G25210*. During the ripening process of fruits, glycan degradation plays a crucial role, particularly in the context of fruit softening. Glycan degradation is a complex biochemical process involving the coordinated action of multiple enzymes, including β-galactosidases. This enzymatic activity results in the weakening of intercellular connections, aligning with our functional analysis that indicates the involvement of *MrBGALs* in functions spanning from intracellular to extracellular processes. Thus, *MrBGALs* plays a significant role in the softening of fruits.

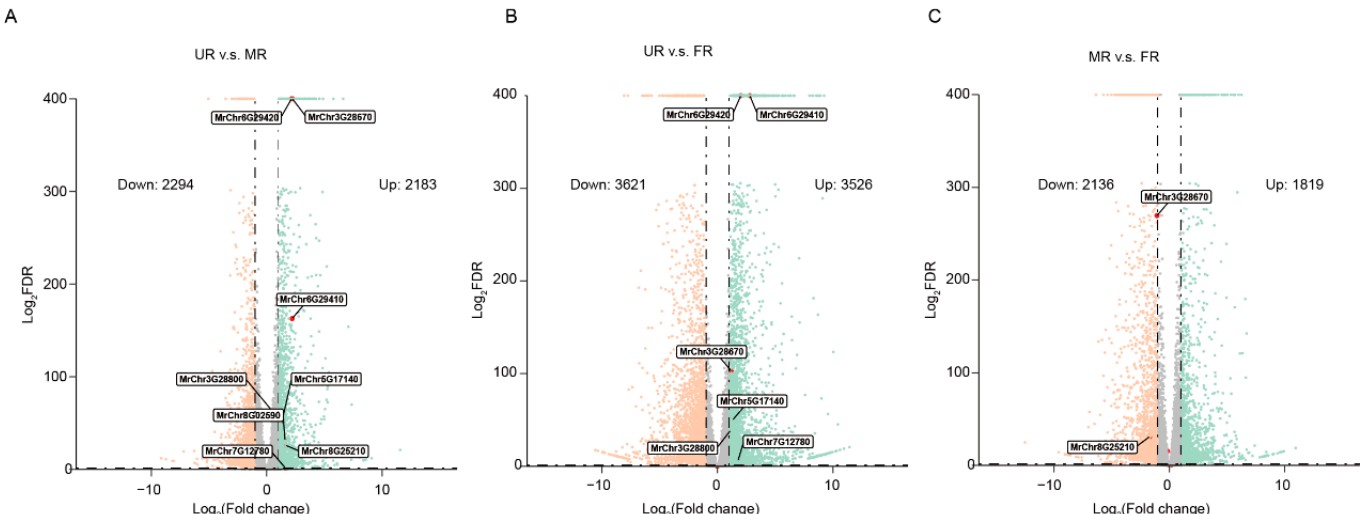

**Figure 5.** *Cont.*

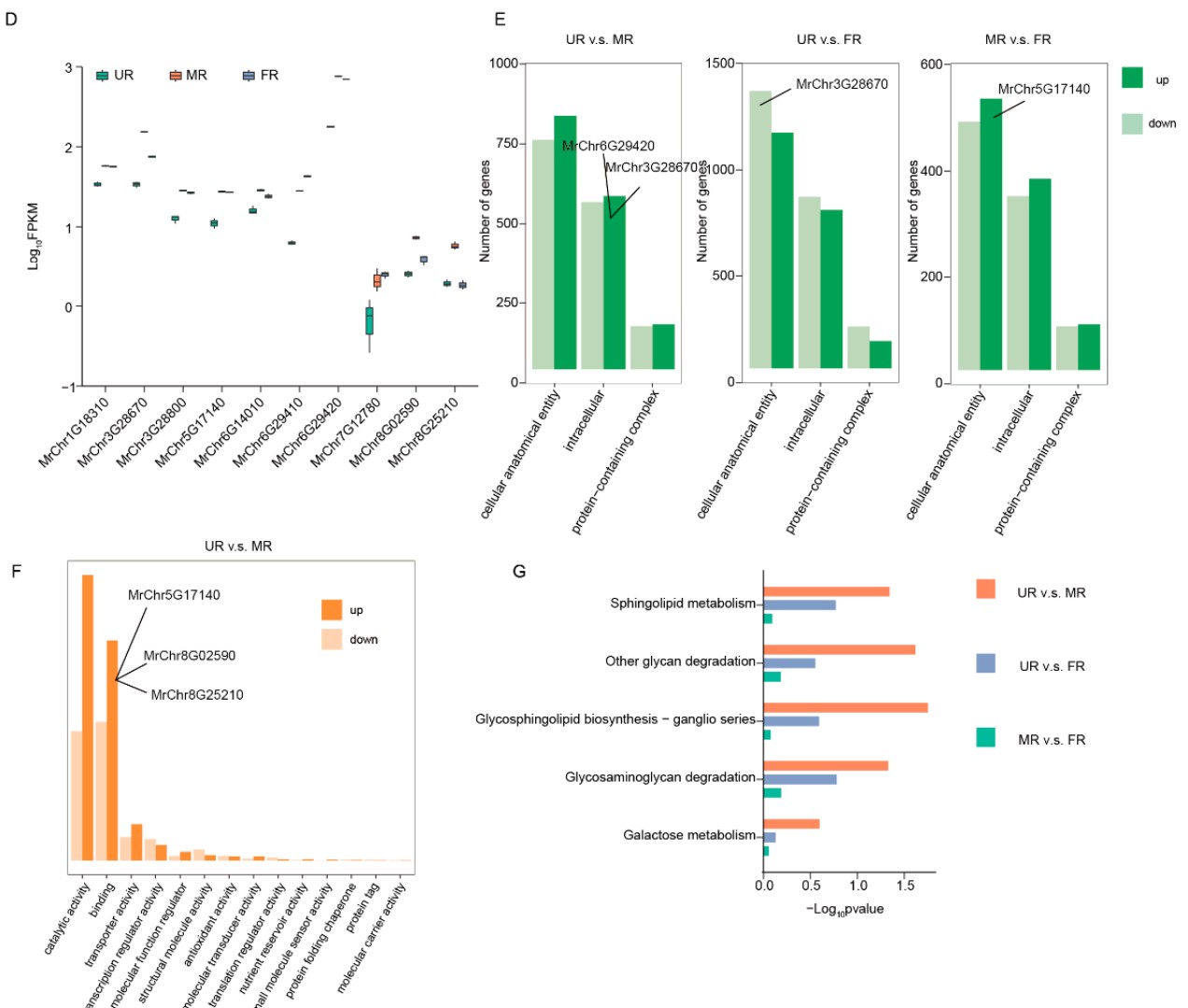

**Figure 5.** Expression patterns of MrBGALs in different fruit development stages. (**A**–**C**) Volcano plots illustrating the differential expression of genes across the three stages. (**D**) Expression patterns specifically highlighting the MrBGALs across these three stages. (**E**) Cell component analysis of gene ontology (GO) enrichment related to differentially expressed genes. (**F**) Molecular function analysis of GO enrichment specifically in the UR-MR stage. (**G**) KEGG (Kyoto Encyclopedia of Genes and Genomes) pathway enrichment related to differentially expressed genes observed across the three stages.

## 4. Discussion

The peculiar morphology of Chinese bayberry fruit, coupled with its gradual softening from its early stages to ripening, makes it imperative to study the textural changes during fruit development. *BGALs* take part in various processes of plant growth and development, particularly in fruit softening. In plants, these enzymes play various roles in different physiological processes, and they are indeed widely distributed. One of the key functions of β-galactosidases in plants is related to the modulation of cell wall polysaccharides [9,10], which can influence plant growth and fruit softening. In this study, we conducted a comprehensive analysis of 15 *MrBGALs* identified through whole-genome exploration of the reference genome (Figure 1A). Our specific focus was on *MrBGALs* that could potentially contribute to fruit development and softening. In apples, BGALs associated with fruit development are categorized into Clade A and Clade B. Our analysis revealed the presence of three *MrBGALs*, namely *MrChr6G29410*, *MrChr6G29420*, and *MrChr3G28670*

in Clade A, and three *MrBGALs* in Clade B, namely *MrChr8G02590*, *MrChr7G12780*, and *MrChr8G25210*. This observation suggests that these *MrBGALs* play a crucial role in fruit development (Figure 1B). In rice, the functional analysis of knockout plants for *OsBGAL9* underscored the pivotal role of *BGALs*. The observed phenotype, marked by short stature and growth retardation, highlights the significance of *OsBGAL9* in the regulation of rice plant growth and development [30]. The conservation of all *MrBGALs* was generally high, with the exception of *MrChr3G28800*, which contained the Glyco_hydro_35 domain motif. Most *MrBGALs* featured GHD and galactose-adhering lectin (Gal_lectin) domains. The expansion of *MrBGALs* at the whole-genome level through gene and motif duplications, with the exception of *MrChr3G28800*, indicated the emergence of new genes during the evolution of *M. rubra* (Figure 2A–C).

In our investigation, PlantCARE was employed to elucidate the transcriptional regulatory mechanisms governing *MrBGALs*. Our findings revealed that the upstream region of *MrBGALs* harbored 13 cis-elements associated with phytohormone responses, including GARE, AuxRE, and ABRE motifs (Figure 3). This suggests a potential regulatory influence of gibberellins, auxins, and abscisic acid on the expression of β-galactosidases. Notably, in sweetpotato, the majority of the *BGALs* were responsive to GA treatment in leaves and stems, indicating a regulatory role of gibberellins in the expression of these genes. Additionally, most *BGALs* were induced by ABA treatment, further emphasizing the involvement of abscisic acid in the modulation of β-galactosidase gene expression in sweetpotato [20]. These observations contribute valuable insights into the hormonal regulation of β-galactosidases, highlighting their responsiveness to specific phytohormones in a context-dependent manner.

In our comparative genome analysis, we identified a gene pair formed by *Atβ-Gal4* (*AT5G56870.1*) and *MrChr6G29420*, suggesting a potential involvement of *MrChr6G29420* in fruit softening (Figure 4). Expression profiling across three fruit development stages reveals the significant importance of *MrBGALs* in fruit development. During the early fruit-ripening stage, the expression levels of the majority of *MrBGALs* increased, whereas in the post-fruit-ripening stage, the expression levels tended to be downregulated. Furthermore, the GO and KEGG enrichment results indicate that *MrBGALs* function from intracellular to extracellular processes and that they play a crucial role in pathways such as glycosphingolipid biosynthesis and glycosaminoglycan degradation. These findings provide robust evidence in support of the pivotal role of *MrBGALs* in both fruit development and the softening process (Figure 5A–G).

These findings contribute valuable insights to the functional study of *MrBGALs* and enhance our understanding of the role of β-galactosidase in fruit softening. The highly conserved nature of most *MrBGALs* and the specific clustering of certain genes in Clades A and B provide a foundation for further investigations into the molecular mechanisms governing fruit development and softening in *M. rubra*. Due to the limited existing research on the softening of *M. rubra* fruit, the identification of the *MrBGALs* in the genome provides an opportunity to investigate the potential participation of numerous genes in this phenomenon. Delving into the features of *MrBGALs*, we illuminate their sequence resemblances and examine the potential consequences of these observations for gaining insights into the mechanism of fruit softening in this distinctive species.

## 5. Conclusions

We systematically identified and characterized 15 *MrBGALs* in *M. rubra*, uncovering their genomic distribution, evolutionary patterns, structural features, and potential roles in plant development. Genomic analysis revealed uneven distribution across chromosomes, with synteny analysis indicating contributions from both whole-genome duplication and tandem duplication events. Evolutionary classification placed *MrBGALs* into seven clades, with specific clades showing potential associations with fruit development. Structural analysis highlighted distinct groups and conserved motifs, while cis-element prediction suggested responsiveness to various environmental stimuli. Comparative genome anal-

ysis with *Arabidopsis thaliana* and *Solanum lycopersicum* identified orthologous gene pairs, including those potentially linked to fruit softening. Expression profiling across different fruit development stages emphasized dynamic regulation, and GO and KEGG analyses provided insights into the molecular functions and pathways associated with *MrBGALs*, particularly during fruit softening. This comprehensive study contributes valuable insights into the multifaceted roles of *MrBGALs* in *M. rubra*, laying the groundwork for future investigations into their specific functions and potential applications in plant development and fruit quality control.

**Supplementary Materials:** The following supporting information can be downloaded at: https://www.mdpi.com/article/10.3390/horticulturae10030225/s1, Table S1: Identification and characterization of MrBGALs; Table S2: Co-linearity analysis of MrBGALs; Table S3: Conserved domain of MrBGALs; Table S4: Cis-element prediction of MrBGALs; Table S5: Cis-element analysis of MrBGALs; Table S6: Co-linearity analysis of MrBGALs and AtBGALs; Table S7: Differential analysis between UR and MR; Table S8: Differential analysis between UR and FR; Table S9: Differential analysis between MR and FR.

**Author Contributions:** L.S. conceived and designed the experiments and wrote the manuscript; Q.Y., S.Z., Z.Y., S.L., X.Z. and H.R. performed the experiments and analyzed the data; X.Q. revised the manuscript. All authors have read and agreed to the published version of the manuscript.

**Funding:** The work was supported by the 'JianbingLingyan' R&D in Zhejiang (2023C02031) and the special breeding program for new varieties in Zhejiang (2021C02066-2).

**Data Availability Statement:** The raw sequencing data presented in this paper were deposited in the Genome Sequence Archive (Accession No. CRA014192) at the National Genomics Data Center, China National Center for Bioinformation/Beijing Institute of Genomics, Chinese Academy of Sciences, and are publicly available at https://ngdc.cncb.ac.cn/gsa, and can be accessed on 24 January 2024. The HiFi and Hi-C data presented in this study are deposited in SRA (http://www.ncbi.nlm.nih.gov/bioproject/937074, accessed on 24 January 2024), and their accession number is PRJNA 937074.

**Conflicts of Interest:** The authors declare that they have no conflicts of interest to report.

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
