# Peer review of "Genome-Wide Identification and Expression Analysis of Beta-Galactosidase Family Members in Chinese Bayberry (Myrica rubra)"

_horticulturae, doi:10.3390/horticulturae10030225_

Round 1

Reviewer 1 Report

Comments and Suggestions for Authors

The presented manuscript devoted to the study of the betta-galactosidase multigene family in Myrica rubra (the betta-galactosidases of M. rubra will be further abbreviated as MrBGALs). During the study, 15 MrBGALs genes were identified in the available M. rubra genome and thoroughly characterized using different bioinformatics approaches. Also, transcriptomic study performed by the authors provided robust evidences for the pivotal role of MrBGALs in both fruit development and the softening process in M. rubra. In my opinion, it is a well written manuscript; it has a good logical structure; all the conclusions are supported by the data. I have read it with interest and joy. However, I suggest for authors to carefully proofread the manuscript. Although generally grammar and style are good there is a strange and awkward places some of which I described below.

Abstract: Please, provide several sentences about betta-galactosidase and its importance in the process of fruit softening. In the current form, this enzyme came out of the blue, and it is not clear why it is significant. Also, put somewhere “betta-galactosidase of M. rubra (MrBGAL)”; all abbreviations in the Abstract must be explained separately from the main text of the manuscript. Consider the following replacement: “Utilizing the phylogenetic tree” -> “Using phylogenetic analysis”.

Introduction: Please, remove the dot after “Chinese bayberry” in the first sentence. Consider the following replacement: “The GH35 conserved site” -> “The GH35 conserved domain”. The last sentence of the second paragraph (“Furthermore, the potential for specific BGAL to function biosynthetically”) is ungrammatical. Consider the following replacement: “β-Galactosidase is believed to expedite fruit softening” -> “The enzyme β-Galactosidase is believed to expedite fruit softening”. Use italic for “β” everywhere in the text. The sentence “Currently, leveraging a specially designed primer design program allows for the delineation of the expression pattern of the BAGL family genes, providing a comprehensive landscape” is awkward, consider to rephrase.

Materials and Methods: Please provide accession-version number of the genome that was used for this work. Consider the following replacement: “RNA-Sequencing Data was analysis” -> “RNA-Sequencing Data was analyzed”.

Results: Consider the following replacement: “To systematically identified and characterized MrBGALs, we extracted the longest transcripts” -> “To systematically identified and characterized MrBGALs, we extracted the longest protein coding sequences”. Please, decipher “WGD” upon the first introduction of the abbreviation; replace “WSD” -> “WGD”. Please, increase the resolution of all figures (be aware, that sometimes the MDPI staff use for the final publication the figure that you put inside the manuscript, rather than the figure that you separately upload into the MDPI web-portal). I do not understand what this sentence try to convey: “To enhance the contextualization of the evolutionary classification, we extracted the longest transcripts and employed the analytical approach previously used for MrBGALs.”; what do you mean by “approach previously used for MrBGALs”? In the Fig3 specify the units of the scale; Is it a Z-scores? In the Fig 4 consider the following replacement: “Heatmap” -> “Collinearity”. Everywhere in the text, Genus name must be fully spelled only at the first mention (i.e. Myrica rubra) and then abbreviated by one letter (i.e. M. rubra).

                I suggest Minor Revision of the manuscript.

Comments on the Quality of English Language

Although generally grammar and style are good there is a strange and awkward places.

Author Response

The presented manuscript devoted to the study of the betta-galactosidase multigene family in Myrica rubra (the betta-galactosidases of M. rubra will be further abbreviated as MrBGALs). During the study, 15 MrBGALs genes were identified in the available M. rubra genome and thoroughly characterized using different bioinformatics approaches. Also, transcriptomic study performed by the authors provided robust evidences for the pivotal role of MrBGALs in both fruit development and the softening process in M. rubra. In my opinion, it is a well written manuscript; it has a good logical structure; all the conclusions are supported by the data. I have read it with interest and joy. However, I suggest for authors to carefully proofread the manuscript. Although generally grammar and style are good there is a strange and awkward places some of which I described below.

Abstract: Please, provide several sentences about betta-galactosidase and its importance in the process of fruit softening. In the current form, this enzyme came out of the blue, and it is not clear why it is significant. Also, put somewhere “betta-galactosidase of M. rubra (MrBGAL)”; all abbreviations in the Abstract must be explained separately from the main text of the manuscript. Consider the following replacement: “Utilizing the phylogenetic tree” -> “Using phylogenetic analysis”.

Response:

Thank you for your valuable feedback. We appreciate your suggestion to provide more clarity on the importance of betta-galactosidase (MrBGAL) in the process of fruit softening, and have revised the abstract accordingly.
“Fruit development and softening play pivotal roles in determining fruit quality and post-harvest shelf life in Chinese bayberry (Myrica rubra). However, the specific role of betta-galactosidase, particularly β-galactosidase of M. rubra (MrBGAL), in facilitating fruit softening remains unclear. In this study, we aimed to address this gap by investigating the involvement of MrBGALs genes in fruit softening.”

We have also ensured that all abbreviations are explained separately from the main text of the manuscript. Additionally, we have incorporated the suggested replacement "Using phylogenetic analysis" instead of "Utilizing the phylogenetic tree" in the appropriate context. Thank you once again for your valuable input.

Introduction: Please, remove the dot after “Chinese bayberry” in the first sentence.

Response:

Thank you for your valuable feedback. We have removed the dot.

Consider the following replacement: “The GH35 conserved site” -> “The GH35 conserved domain”.

Response:

Thank you for your valuable feedback. We have revised the sentence accordingly.

The last sentence of the second paragraph (“Furthermore, the potential for specific BGAL to function biosynthetically”) is ungrammatical. Consider the following replacement: “β-Galactosidase is believed to expedite fruit softening” -> “The enzyme β-Galactosidase is believed to expedite fruit softening”.

Response:

Thank you for your valuable feedback. We have revised the sentence accordingly.

Use italic for “β” everywhere in the text.

Response:

Thank you for your valuable feedback. We have revised the β in the main text.

The sentence “Currently, leveraging a specially designed primer design program allows for the delineation of the expression pattern of the BAGL family genes, providing a comprehensive landscape” is awkward, consider to rephrase.

Response:

Thank you for your valuable feedback. We have revised the sentence accordingly.

“The use of a specialized universal primer facilitates a comprehensive understanding of the expression patterns of the BAGL family genes”

Materials and Methods: Please provide accession-version number of the genome that was used for this work. Consider the following replacement: “RNA-Sequencing Data was analysis” -> “RNA-Sequencing Data was analyzed”.

Response:

Thank you for your valuable feedback. We have revised the sentence accordingly.

Results: Consider the following replacement: “To systematically identified and characterized MrBGALs, we extracted the longest transcripts” -> “To systematically identified and characterized MrBGALs, we extracted the longest protein coding sequences”.

Response:

Thank you for your valuable feedback. We have revised the sentence accordingly.

Please, decipher “WGD” upon the first introduction of the abbreviation; replace “WSD” -> “WGD”.

Response:

Thank you for your valuable feedback. We have made the necessary changes.

“The results revealed that 5 gene pairs from 15 MrBGALs appeared to have arisen from Whole-Genome Duplication (WGD) or segmental duplications.”

Please, increase the resolution of all figures (be aware, that sometimes the MDPI staff use for the final publication the figure that you put inside the manuscript, rather than the figure that you separately upload into the MDPI web-portal).

Response:

Thank you for bringing this to our attention. All figures provided in the manuscript are of high resolution (300dpi) and in RGB format. Uploading the figures with .tiff and .ai separately should help ensure that the editorial office can access the higher resolution versions for publication. If there are any further instructions or specific requirements regarding the figures, please let me know.

I do not understand what this sentence try to convey: “To enhance the contextualization of the evolutionary classification, we extracted the longest transcripts and employed the analytical approach previously used for MrBGALs.”; what do you mean by “approach previously used for MrBGALs”?

Response:

Thank you for bringing this to our attention. We apologize for the confusion. We will revise the sentence to clarify the analytical approach used for the evolutionary classification.

“To enhance the contextualization of the evolutionary classification, we applied the analytical approach previously used for the identification of MrBGALs to extract the longest transcripts from the selected six species for evolutionary analysis.”

In the Fig3 specify the units of the scale; Is it a Z-scores?

Response:

Thank you for bringing this to our attention. We apologize for the confusion. The scale is Z-score. We have made the necessary changes.

In the Fig 4 consider the following replacement: “Heatmap” -> “Collinearity”.

Response:

Thank you for bringing this to our attention. We have made the necessary changes.

Everywhere in the text, Genus name must be fully spelled only at the first mention (i.e. Myrica rubra) and then abbreviated by one letter (i.e. M. rubra).

Response:

Thank you for bringing this to our attention. We have made the necessary changes.

Thank you for your patience and valuable feedback; it will greatly contribute to improving and refining our research.

Reviewer 2 Report

Comments and Suggestions for Authors

My main concerns are with the presentation of the data in section 3.6. Details of it and minor comments are highlighted in the attached file.

Comments on the Quality of English Language

Minor editing of English language required

Author Response

My main concerns are with the presentation of the data in section 3.6. Details of it and minor comments are highlighted in the attached file.

Response:

Thank you for your kind words and for acknowledging the helpfulness of the suggestions and revised sentences. We have revised the sentence accordingly to reflect your gratitude. Additionally, we have reviewed the comments provided in the attached file regarding the presentation of the data in section 3.6 and made the necessary revisions to address the concerns raised. If there are any specific details or minor comments that require further attention, please do not hesitate to let us know, and we will ensure to address them promptly.
